developmental biology, evolution, genetics

balancing selection, oysters, genetics, poolseq, larval development

**Author for correspondence:**
Evan Durland
e-mail: durlandevan@gmail.com

# Temporally balanced selection during development of larval Pacific oysters (*Crassostrea gigas*) inherently preserves genetic diversity within offspring

Evan Durland[1,2], Pierre De Wit[2] and Chris Langdon[1]

[1]Department of Fisheries and Wildlife and Coastal Oregon Marine Experiment Station, Hatfield Marine Science Center, Oregon State University, Newport, OR 97365, USA
[2]Department of Marine Sciences, Tjärnö Marine Laboratory, University of Gothenburg, Strömstad, Sweden

ED, 0000-0003-1322-6810; PD, 0000-0003-4709-3438

Balancing selection is one of the mechanisms which has been proposed to explain the maintenance of genetic diversity in species across generations. For species with large populations and complex life histories, however, heterogeneous selection pressures may create a scenario in which the net effects of selection are balanced across developmental stages. With replicated cultures and a pooled sequencing approach, we show that genotype-dependent mortality in larvae of the Pacific oyster (*Crassostrea gigas*) is largely temporally dynamic and inconsistently in favour of a single genotype or allelic variant at each locus. Overall, the patterns of genetic change we observe to be taking place are more complex than what would be expected under classical examples of additive or dominant genetic interactions. They are also not easily explained by our current understanding of the effects of genetic load. Collectively, temporally heterogeneous selection pressures across different larval developmental stages may act to maintain genetic diversity, while also inherently sheltering genetic load within oyster populations.

## 1. Introduction

Balancing selection is a form of natural selection that maintains multiple genotypes or traits in a population across generations. Traditionally, this form of natural selection has been explained primarily as a result of genotypic over-dominance, frequency-dependent selection or adaptation of sub-populations with limited genetic connectivity to the larger whole [1]. Molecular evidence of balancing selection for alleles and genotypes has also been reported as a result of competition between pathogens and their hosts [2], or a product of environments that are temporally variable between generations [3]. The evolutionary importance of balancing selection lies in its role in maintaining genetic diversity in populations, even in those faced with intense selection pressures against alleles and phenotypes in discrete locations or periods of time.

The polygenic nature of many fitness traits, however, complicates straightforward identification of balancing selection with molecular genetic markers [1]. Additionally, pleiotropy, epistasis and complex genomic architectures may also create unexpected outcomes for genes putatively affected by balancing selection, as well as alleles at linked loci which may be deleterious to the organism [4]. The myriad processes which influence the persistence of alleles and genotypes may simultaneously affect the strength of directional or balancing selection for discrete loci as well as the ability of molecular tools to detect the signals of meaningful allele frequency change underpinning these selective pressures. For example, traditional theories regarding the persistence of mutations with negative phenotypic effects in a population, termed 'genetic

load', suggest that there exists a balance between acquiring disadvantageous mutations and purging these alleles through natural selection [5]. Balancing selection, however, may potentially 'shelter' genetic load in some organisms, through overdominance or by the association of the deleterious locus or loci with additional genes which are under opposing selective pressures [4].

Balancing selection, whether arising via overdominance in stable environments or alternating selection pressures in variable ones, has previously been presented as the result of selective forces acting across multiple generations [3,6–9]. Many species, however, have complex life histories with distinct physiological stages and genetic selective pressures are unlikely to be uniform across all of them. Complex developmental processes may represent a morphologically heterogeneous 'landscape' which is analogous to examples of balancing selection occurring in spatially variable habitats [10]. Developmental transcriptomes of a variety of invertebrates [11–15] demonstrate that the large and diverse genetic networks, that underpin physiological changes taking place early in the life of these organisms, are not uniformly profiled as a linear sequence of gene expressions. Many hundreds or thousands of genes exhibit stage-specific [14], or even oscillatory, expression patterns throughout development [16]. This allows for the possibility that an allele or genotype which provides improved fitness during one phase of development may not have a consistent effect at another, leading to temporally offset selection pressures which may result in a form of balancing selection across the entire developmental period. Previous investigations have evaluated the genetic diversity of developmentally critical genes [17] and their patterns of expression during early life stages in model species [18]; however, direct measurements of changes in allele frequencies in a composite population throughout development are lacking, particularly in non-model organisms. Some changes in allele frequencies during development are likely to be simply explained by directional selection. Other genetic changes, on the other hand, may reflect stage-specific or varying patterns of gene expression that are 'temporally balanced' across development.

In this study, we describe genetic changes taking place within a single generation of a population of Pacific oysters (*Crassostrea gigas*) over 22 days of larval development. This species is characterized by having high-genetic diversity [19,20] but also a substantial genetic load, leading to low survivorship during larval development [21]. The early life stages of oysters are morphologically complex, transitioning from embryo to shelled 'D-hinge' larvae within 24 h of fertilization, followed by growth and development of a planktotrophic, motile 'veliger' stage over approximately 12–16 days, leading to pre-metamorphic 'pediveliger' larvae. At this stage, pediveligers attach themselves to a substrate and undergo metamorphosis to become sedentary juvenile 'spat' (a process alternatively referred to as 'settlement') which takes place over a period of several hours under natural conditions. Developmental transitions of larvae also tend to coincide with periods of elevated rates of mortality, as well as pronounced changes in genetic composition [22].

In order to investigate temporal patterns of genotype-dependent mortality during larval development, we created a composite pool of oyster larvae, consisting of 95 individually fertilized crosses from a naturally breeding population in the Pacific northwest, USA. We reared them from fertilization to settlement under standard hatchery conditions and sampled larvae from each of five replicate cultures throughout development for pooled DNA analyses. Using 751 single nucleotide polymorphisms (SNPs) distributed across the genome, we conducted parametric statistical tests and applied k-means clustering to distinguish patterns of change as a result of genotype-dependent mortality. We also developed a predictive algorithm to model how changes in allele frequency translate to shifts in genotype composition and fitness. To our knowledge, this is the first time that pooled allele frequency data have been used to model genotype frequency changes over development in any organism. Our results indicate that a combination of genetic load and temporally balanced selection across larval developmental stages best explain the observed patterns.

## 2. Methods

Comprehensive details regarding all culture methods (broodstock conditioning, cross design, larval culture and sampling) can be found in our previous related study [23] but these will be briefly reviewed here.

### (a) Broodstock conditioning and cross design
In the spring of 2015, approximately 60 wild oysters previously obtained from Willapa Bay, WA, were brought into conditioning tanks at the Hatfield Marine Science Center (HMSC), Newport, Oregon. Broodstock was continuously provided with flowing seawater and ample microalgal diets for *ca* six weeks, while seawater temperature was gradually increased from ambient (approx. 11°C) to 20°C. In June, 19 female and five male oysters were individually paired in a fully factorial mating design (every male paired with every female), to create 95 full-sibling (1 female × 1 male) crosses. Fertilized eggs were proportionally combined to form a composite embryo pool that contained approximately equal proportions of each of the 95 crosses.

### (b) Culture units and water quality
Larvae were reared in 10 l polycarbonate chambers (BearVault, San Diego, CA) fitted with a lid and sealed with a silicone ring (McMaster-Carr, Santa Fe Springs, CA). Units were filled with hatchery seawater (25°C, salinity of 32, 10 µm-filtered) that had been aerated overnight to equilibrate dissolved $pCO_2$ to ambient levels (approx. 400 µatm $CO_2$, pH = 7.9–8.1, $\Omega_{arag}$ = 2.3–2.7 [23]). The pH, temperature and dissolved oxygen of individual culture chambers were monitored daily and no substantial aberrations in water quality were observed [23].

### (c) Larval culture
Approximately 5 h after fertilization about 200 000 embryos from the pool were distributed to each of five culture replicates at an effective stocking density of 20 larvae ml$^{-1}$. Culture water was changed every 48 h by sieving oyster larvae on a mesh screen, filling the culture units with fresh seawater and re-stocking the larvae. Importantly, conservative screen sizes were used in order to retain all surviving larvae, with no selection for growth (electronic supplementary material, table S1). Mixed microalgal diets of *Isochrysis galbana* and *Chaetoceros gracilis* were supplied once daily, starting 2 days post-fertilization (dpf) at a concentration of 20 000 cells ml$^{-1}$ that was subsequently increased by 5000 cells ml$^{-1}$ d$^{-1}$. Larval densities were reduced to 10 ml$^{-1}$ on day 2, 5 ml$^{-1}$ on day 6 and 1 ml$^{-1}$ at the pediveliger stage on day 14 to provide optimal densities for larval growth and survival. Larval density was reduced with no selection for larval size. After the

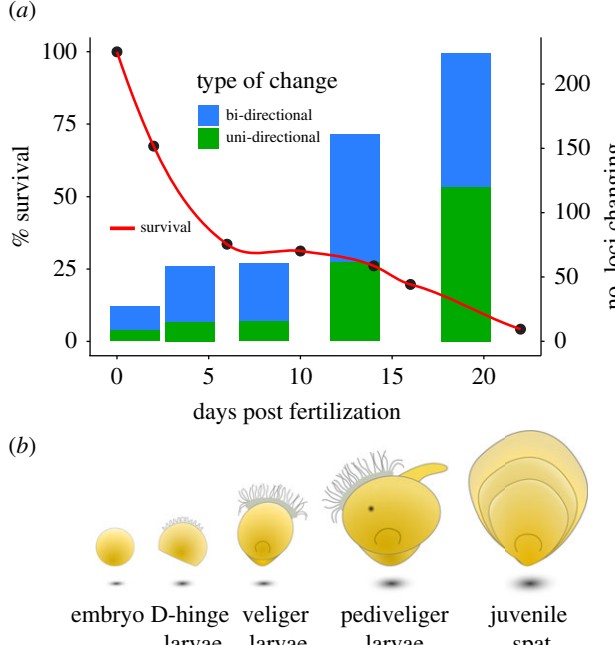

**Figure 1.** (*a*) Temporal patterns of mortality and significant changes in mean allele frequency across larval development. The points and line represent mean cumulative mortality across the 22-day culture period, bars represent bidirectional and uni-directional changes in allele frequencies at each time interval (non-cumulative). (*b*) Five general stages of development (embryo, D-hinge, veliger, pediveliger and spat) are depicted (not to scale) on the bottom row, relative to the developmental time in (*a*). (Online version in colour.)

appearance of eyed larvae (indicating a readiness to metamorphose), larvae were screened on a 240 µm sieve to retain pediveliger larvae which were subsequently induced to metamorphose by exposure to $1.8 \times 10^{-4}$ M epinephrine for 2 h [24] and then returned to the culture unit. Metamorphosis was induced in this fashion on days 16, 18 and 20. The experiment was terminated on day 22 after a majority of the eyed larvae had metamorphosed ($\bar{x} = 62\%$).

## (d) Larval sampling
Larvae were sampled and counted in each culture unit on days 2, 6, 10, 14, 16 and 22. Survival estimates accounted for larvae removed for sampling and for adjustments in densities to obtain a cumulative survival estimate across the entire experimental period (figure 1*a*). Egg and larval samples for DNA extraction contained approximately 200–3000 individual oysters (depending on age) per culture unit (electronic supplementary material, table S1). Initial samples of fertilized eggs were taken approximately 1 h after fertilization, after the appearance of a polar body. Samples on day 22 were a composite of a metamorphosed unattached juvenile spat as well as eyed larvae that were retained on a 240 µm screen. From each sample, genomic DNA was extracted using a CTAB extraction method with RNAse treatment [25]. DNA concentration and purity were assessed using a Qubit fluorometer (Thermo Scientific). The 2bRAD libraries were prepared following the established protocol [26], using the BcgI restriction enzyme. All individual samples ($n = 26$; egg pool + 5 replicates × 5 time points) were given unique barcodes and pooled (with samples from a different study) in sets of approximately 32 samples per sequencing lane ($n = 5$). Single-read, 50 base-pair (bp) target length sequencing was conducted on an Illumina HiSeq2500 platform at the SNP&SEQ Technology Platform at The Swedish National Genomics Infrastructure, Uppsala.

## (e) Bioinformatics
The bioinformatic analysis of DNA sequences followed the pipeline constructed by Dr Eli Meyer (v. 3.0; scripts and manual available at http://eli-meyer.github.io/2bRAD_utilities/index.html). Briefly, de-duplexed raw reads were truncated to 36 bp in length and quality filtered for reads that had less than 10 bp with Phred quality scores less than 20. Reads were then filtered for adaptor sequence contaminants using a kmer size of 12 [27]. Cleaned, high-quality reads were mapped to a reference genome [20] using the SHRiMP software package [28] with default mapping parameters but allowing for a maximum of three genomic alignments per read and retaining the single best alignment. Alignments were then filtered to retain only those with greater than or equal to 30 bp matching the reference sequence. Sequence alignment map files were converted to tab-delimited files with read counts of each nucleotide at each locus and merged. A custom Python script was used to then remove reads with a sequencing depth of less than 50 or greater than 1000 [29] and loci with greater than 2 alleles detected. 'Minor' alleles were designated as the less abundant of the two alleles for each SNP, averaged across all samples. Lastly, polymorphic loci were filtered to one SNP per 36 bp tag (i.e. tags with greater than 1 SNP were excluded) to limit complications arising from potential ambiguities in the mapping of the reads which may erroneously result in the detection of multiple polymorphisms in close proximity to one another. Two samples (one from day 6 and one from day 16) received low sequencing coverage and did not pass the above filtering steps. Downstream analyses proceeded with the remaining $n = 24$ samples. Bioinformatic methods and custom scripts can be found at https://github.com/E-Durland/oyster-poolseq, and quality filtering and mapping statistics can be found in the electronic supplementary material, information. All bioinformatics analyses were conducted on computing infrastructure at the Center for Genomic Research and Biocomputing Core Laboratories at Oregon State University.

## (f) Data analysis
### (i) Detecting change in allele frequencies
A total of 5373 SNPs were retained from the filtering steps (described above) for statistical analysis. Loci in this dataset were filtered for missing data and rare polymorphisms by keeping only loci for which three or more replicates (cultures) per time point had reads present. Loci were filtered for minor allele frequency (MAF) greater than 1% in the egg pool. After these steps, there were 751 SNPs left for statistical modelling. The read count data for each allele at each remaining locus were analysed with a binomial generalized linear model using the formula: $(A_1 : A_2) \sim \beta_{\text{Age}}$ where $A_1$ and $A_2$ refer to the read counts of alternative alleles at each locus and $\beta_{\text{Age}}$ is the parameter estimate for the age of the sample (0 to 22 dpf). Age was incorporated into the model as a multi-level factor rather than a continuous variable because the predominant type of genetic change we observed was variable and nonlinear between ages. For each locus, the $p$-value for 'age' effects (type I sum of squares) was adjusted with the Benjamini–Hochberg procedure for 5% false-discovery rate and adjusted $p$-values were deemed significant at $p < 0.05$. All statistical procedures were conducted in R (v. 3.6.1) [30].

## (g) Parametric tests of significant change
In order to investigate the dynamic patterns of change in allele frequencies between time points for loci with significant 'age' effects, we used pairwise multiple comparison tests (Tukey's HSD) for mean allele frequencies at sequential time points. This method stringently detects significant and abrupt changes in allele frequencies between two time points while controlling for

false-discovery inflation from multiple comparisons within the same locus. We categorized significant changes (or lack thereof) sequentially between sampling points and categorized them as: (i) 'gradual', (ii) 'uni-directional' (UD), or (iii) 'bi-directional' (BD). 'Gradual' changes (G) were assigned to loci with insignificant ($p > 0.05$) pairwise differences between subsequent time points. UD changes were identified by significant changes in allele frequencies between consecutive time points (e.g. day 2 to day 6) which were solitary or uniform in direction (up or down). Loci that had two or more significant changes in allele frequencies in opposite directions (e.g. up, then down or reverse) were categorized as BD. Example trajectories in mean allele frequencies for each category, and the derived change in allele frequencies ($\Delta$AF) between each sampling interval, are represented in the electronic supplementary material, figure S1.

## (h) Clustering allele frequency trajectories

k-means clustering was used to group locus-specific patterns of $\Delta$AF between sampling points, similar to methods used with time-series gene expression data [31]. First, we determined the optimal number of clusters ($k = 5$) by testing multiple indices on a k-range of 1–20 with a Manhattan dissimilarity matrix using the NbClust package in R [32]. We then executed the clustering algorithm ('kmeans' in R) to group each locus into a cluster for further interpretation. A visualization of the clustering results can be found in the electronic supplementary material, figure S2. Mean allele frequencies for all loci and time points, along with corresponding p-values, parametric categories, cluster assignments and linkage mapping locations can be found in the electronic supplementary material, file S1.

## (i) Mapping markers to linkage groups

In order to assess the assumption of independence of loci, we assigned SNPs to linkage groups (LGs) by comparing genomic mapping locations to markers on a previously published linkage map [33]. To accomplish this, we used a reference genome [20] as a 'bridge' to the linkage map. For each SNP in our dataset that shared a genomic scaffold with a 'mapped marker' from the linkage map (1 SNP: 1 mapped), we assigned it the same position as the reported mapped marker. If multiple SNPs from our dataset were found on a scaffold that was represented by a single mapped marker (greater than equal to 2 SNPs: 1 mapped) all the corresponding SNPs from our dataset were assigned to the same genomic position of the mapped marker. When multiple mapped markers were found on the same scaffold, SNPs in our dataset were assigned the linkage position of the marker that was nearest in the scaffold (by base-pair location). Loci existing on genomic scaffolds which were not found on the linkage map or scaffolds which appeared on multiple LGs were omitted from this step of the analysis. In total, $n = 200$ markers (approx. 27%) were mapped to LGs with a mean distance between markers of approximately 6.4 cm ($\pm 5.77$ s.d. among LGs; see the electronic supplementary material, files S2 and S3).

## (j) Estimating genotype proportions from minor allele frequencies and population structure

One of the inherent limitations of pooling individuals for DNA sequencing is the inability to infer genotypes from allele frequency data alone. In this study, however, the larval pool was not a random sample from a larger population but the result of a controlled factorial cross design (5 male × 19 female) with the allele frequency of the fertilized eggs reflecting the complete gamete pool of the parental oysters. For each locus, we used the MAF in the egg pool as the basis for a simulation of $n = 50$ broodstock populations with the same proportion of major and

minor alleles randomly distributed among them. For each simulated population, we then randomly assigned 5 'males' and 19 'females' and created a 5 × 19 cross *in silico*. Each simulation predicted the corresponding genotype composition of the fertilized eggs based on Mendelian ratios and presuming an absence of distortions from the meiotic drive or fertilization success [21,34]. Despite the unbalanced cross design in this case, genotype ratios in fertilized eggs were largely consistent with expectations under Hardy–Weinberg equilibrium (HWE; electronic supplementary material, figure S3). A full description and demonstration of this simulation method can be found at https://github.com/E-Durland/Genotype_simulator.

Equipped with only a time-series dataset of survival and MAF, it is impossible to calculate changes in genotype frequencies, even if the initial composition can be estimated (as above). The range of possible trajectories (time-series changes) for each genotype at a locus (AA/AB/BB), however, is not limitless and is bounded by empirical observations of the number of larvae surviving and the MAF at that time. We used these empirical 'boundaries' to direct the simulation over a range of trajectories for each of the three genotypes (AA/AB/BB) at each locus independently. In order to do this, we developed an algorithm to simulate trajectories through iterative randomization of the modified Hardy–Weinberg fitness equation:

$$\bar{w} = p^2 w_{AA} + 2pq w_{AB} + q^2 w_{BB},$$

where $p$ and $q$ represent the fractional abundance of major and minor alleles, respectively, $w_{AA}/w_{AB}/w_{BB}$ are the relative fitness of each genotype and $\bar{w}$ represents mean fitness of the group. The model we developed randomly assigned fitness values (0–1) for each of the three genotypes and calculated the resulting changes in larval survival (by genotype) and MAF in the simulated population. At each time point (days 2, 6, 10, 14 and 22), the simulation was then compared to corresponding empirical estimates of survival and MAF (means among replicates) and discarded if it deviated by more than 10% from either. A 'successful' simulation then proceeded to the next time point and repeated the randomization. Each full iteration of this process simulated one of the potentially numerous ways that genotype frequencies for a single locus could have changed in the larval population across development while still accounting for measurements of mortality and changes in MAF. Repeated simulations allowed for the estimation of mean frequency and fitness for each genotype at each time point for each locus. An example simulation of a single locus is visualized in the electronic supplementary material, figure S4.

We tested this simulation method against data from a previous study that evaluated genotypic changes of oyster larvae during development with a mixed marker set [21]. Using a set of 12 candidate markers from that study, sampled at various time points through larval development, we produced 1200 simulations for 186 genotype frequencies (electronic supplementary material, figure S5). Empirical estimates fell within 95% confidence intervals of simulations greater than 90% of the time ($n = 176$) and deviated from mean simulated frequencies by less than 10% in more than 80% of the calls ($n = 153$; electronic supplementary material, figures S6 and S7). The model and data used for testing it can be found at: https://github.com/E-Durland/Genotype_forecaster.

## (k) Analysis of genotype distortions

In order to rationalize how changes in allele frequency from fertilization to settlement translated to shifts in the composition of each of the three genotypes at a locus (AA/AB/BB), we compared two approaches: (i) changes relative to genotype frequencies in the fertilized egg pool, and (ii) relationship between HWE estimates from the MAF in the spat pool to

predicted genotype frequencies at the same stage. We used linear models and compared the goodness-of-fit based on F-statistics, Akaike and Bayesian information criterion scores (electronic supplementary material, table S2). All models had highly significant $p$-values for their respective parameter ($p \ll 0.01$) but HWE estimates from the MAF in the spat pool were more strongly supported (see residual plots in the electronic supplementary material, figure S8) and, hence, used for interpretation. Prediction intervals for the linear models across the range of MAF in the spat pool can be found in the electronic supplementary material, figure S9.

## 3. Results and discussion

### (a) High larval mortality occurs with abundant, dynamic genetic changes

Total mortality in the larval oyster population was high with an average of only approximately 4% of spat and eyed larvae remaining at the end of the 22-day culture period. Extensive mortality during early life stages (type III survivorship), however, is common for Pacific oysters [21] where there are markedly greater losses occurring early and late in the development period (figure 1a). This profound winnowing of the larval population was accompanied by numerous changes in allele frequencies within the population. Among the 751 SNPs which passed filtering steps, 473 (approx. 63%) exhibited significantly distorted (altered) mean allele frequencies at one or more time points during larval development. Parametric tests comparing changes between specific time points indicated that 139 loci (approx. 29% of those that changed) showed 'gradual' changes in allele frequency; 207 (approx. 44%) were temporally distinct and UD; and 127 (approx. 27%) had alternating, or BD patterns of change. From a developmental perspective, later larval stages, including metamorphosis (days 10–22), displayed more than twice as many loci undergoing significant changes in allele frequency ($n = 385$) when compared to early developing larvae (days 0–10; $n = 146$; figure 1). This finding is in keeping with previous estimates regarding the timing of the 'expression' of genetic load during larval oyster development [22,35].

Using k-means clustering, we were able to classify patterns of allele frequency changes over development in these larval populations. Each locus (SNP) was clustered into a group with other SNPs that shared their overall pattern of change in mean allele frequencies between time points (figure 2). Clusters were ordered by descending membership (number of loci) which, incidentally, also placed them in increasing levels of dynamic change. There was a general agreement between the conservative but coarse classifications from parametric tests and the non-parametric cluster assignments (electronic supplementary material, table S3): 'gradual' (G) and UD SNPs were primarily assigned to clusters with low levels of apparent change (clusters 1 and 2) and BD SNPs were over-abundant in clusters exhibiting the most dramatic allele frequency trajectories across development (clusters 4 and 5). Quantitatively, the majority of loci could be assigned to clusters with low-orders of change (clusters 1 and 2, $n = 362$ or approximately 77%) but nearly a quarter of loci had higher order patterns of change during the 22-day larval development period (clusters 3–5, $n = 111$ or approx. 23%).

Abundant genotypic distortions in juvenile oysters, relative to combinations of parental genotypes, have been reported previously [21,22,35]. These studies suggested that mutations with negative phenotypic effects (genetic load), present in both domesticated and naturally recruiting stocks of C. gigas, render greater than 95% of oyster larvae genetically inviable. The high-genetic load in Pacific oysters is surprising because deleterious mutations theoretically should be subjected to intense selection pressures and be removed from open breeding populations [5]. It has been proposed that a very high mutation rate in oysters (approx. 90x that of Drosophila [21]) is one way to account for the persistence of deleterious mutations in this species. Under this hypothesis, the traditional mechanisms of purging are inadequate in reducing the overall genetic load (or to achieve allele frequency equilibrium) owing to the rapid production of new deleterious alleles. This is supported by evidence from randomly bred progeny from naturalized C. gigas that had 11–19 detrimental alleles unique to each full-sibling family [21]. Consequentially, in the current study, it is unlikely that the overall observed magnitude of genetic change during larval development is significantly skewed by founder effects or the artificial inflation in the frequencies of rare variants from male oysters with disproportionate contributions to the larval pool (5 males × 19 females). In essence, genetic load in the Pacific oyster may not be definable by a discrete set of genetic markers, but instead by a constantly shifting pool of new alleles that arise in each generation and have significant impacts upon larval survival.

Compared to previous investigations of genotype-dependent mortality in larval oysters, this study uses a substantially larger set of genetic markers with much greater temporal resolution. The results indicate that genetic changes occurring during larval development in oysters are not only significant but more plentiful and complex than previously understood. When temporally significant changes in allele frequency were compared to 'overall' shifts in MAF between day 2 (the first day with replicated measurements) and day 22, less than half ($n = 181$; approx. 38%) of all measured loci that were changing during development had 'final' allele frequencies that were significantly distorted at the end of the developmental period (figure 3). For the remaining approx. 62% of loci, significant changes in allele frequencies at one stage during larval development were offset by additional changes in the opposite direction, balancing the overall distortion in allele frequencies across larval development.

The pattern of balanced changes in allele frequencies appears to occur across numerous regions of the genome. There is little evidence to suggest that multiple SNPs within each cluster share genomic proximity (i.e. are non-independent) that would artificially inflate the number of SNPs with similar patterns of change. The mean distance for mapped markers within each cluster range from approximately 14–28 cm with no clear groupings in discrete regions (see the electronic supplementary material, table S4 and files S4–S8). It seems that, for the Pacific oyster, genotype-dependent mortality is a genome-wide phenomenon.

### (b) Dynamic changes imply temporally variable fitness of genotypes

In order to estimate how observed changes in allele frequencies relate to temporal patterns of change in genotype composition and relative fitness, we developed an iterative algorithm to model these trajectories across larval development. For loci with significant changes in MAF ($n = 473$),

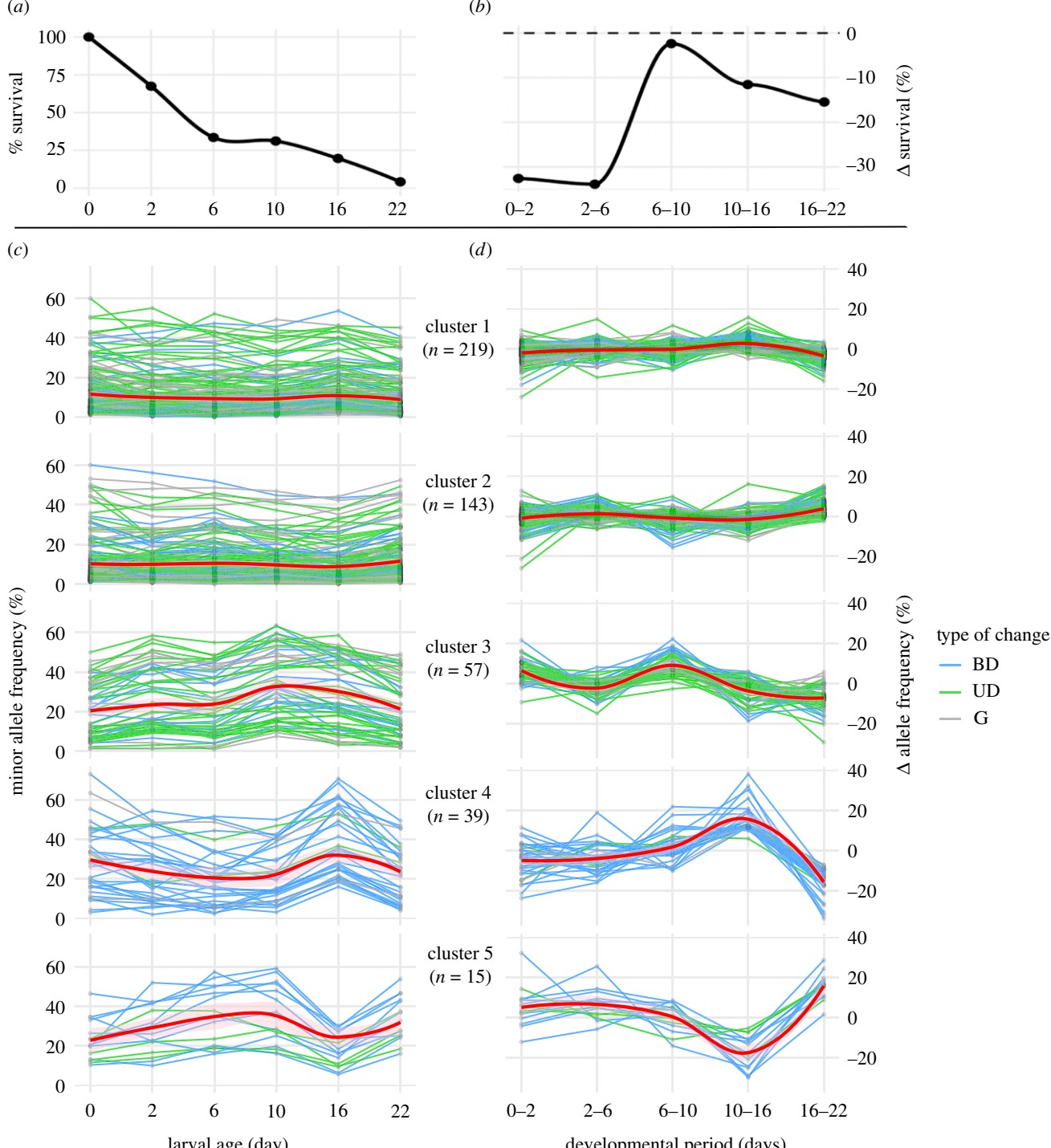

**Figure 2.** Change in allele frequency corresponding with larval mortality. (a) Mean survival of larvae in replicate cultures from fertilization to settlement (day 22). (b) Change (Δ) in survival between sampling periods. (c) Clustered trajectories of minor allele frequencies (n = 473 in total). (d) Change (Δ) in MAF between sampling points. Line colours correspond to patterns of change identified by sequential pairwise parametric tests (G, gradual; UD, uni-directional; BD, bi-directional). Bold red lines are localized estimate (loess) trajectories for all SNPs within a cluster (pink ribbons ± s.d.). (Online version in colour.)

we averaged values among $n = 50$ simulations for each of six time points ($n = 141\,900$ simulations in total). We then modelled mean trajectories of relative fitness values for each of the three possible genotypes across development and grouped these trajectories into previously defined clusters (figure 4). Fitness values, rather than genotypic proportions, were used because they are un-skewed by starting allele frequency (and, thus, genotype composition) and are more informative with regard to selection processes. Although modelled trajectories are locus-specific, the aggregate trends within each cluster demonstrated a surprising

amount of temporal heterogeneity in relative fitness for genotypes across larval development. For SNPs in clusters with low-order change (1 and 2), the aggregate trend supports an additive genetic hypothesis: homozygotes with the major allele (AA) displayed a somewhat consistently greater fitness than heterozygotes (AB) which, in turn, fared generally better than minor homozygotes (BB). For those in clusters with higher order levels of change (3–5), however, simple explanations of additive or dominant genetic interactions do not adequately address the temporal changes we observed. It can be argued that dominance broadly explains

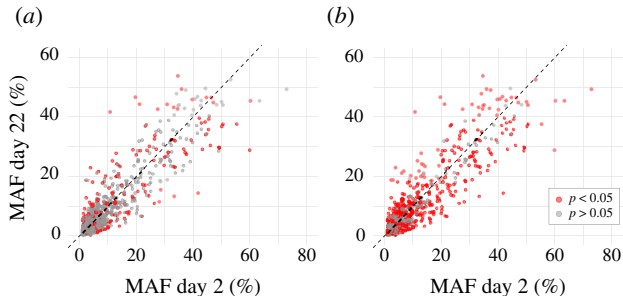

**Figure 3.** Overall changes in MAF between days 2 and 22 post-fertilization. (*a*) Loci with overall distortions were determined significant (in red) by pairwise comparison between day 2 and 22 (*n* = 181). (*b*) Loci with temporally significant distortions (in red) at one or more time points in development (*n* = 473). Each point represents a single locus (SNP). Distance from the dashed centre line is the overall distortion of allele frequency between day 2 and 22. (Online version in colour.)

the relatively improved fitness of AA and AB genotypes for the first approximately 10 days of larval development, but significant switches in fitness estimates from day 10 to 22 suggest that additional mechanisms (such as overdominance and a switch in favourability between A and B alleles) drove changes at the same loci during the final developmental stages.

Metamorphosis is a major larval developmental transition involving a host of structural, biochemical and genetic changes [35–37] fuelled by catabolism of energy reserves [38,39]. The reversal of fitness trajectories for genotypes at this stage, as shown by the aggregate trends for clusters 2–5, indicates that many favourable alleles and genotypes during earlier veliger larval stages had opposite effects on metamorphosis and settlement. It cannot be determined with these data whether this reversal was owing to direct negative effects on metamorphosis itself, such as structural re-organization, or an acute manifestation of accumulated chronic, sub-lethal effects on general larval fitness, such as feeding, metabolism and accumulation of lipid reserves. Further investigations are needed to untangle the host of possible genetic factors affecting settlement success.

## (c) Temporally balancing selection favours heterozygotes

From the standpoint of population genetics, it is important not only to parameterize the total change in allele frequencies at various loci but also the shifts in genotypic proportions that accompany them. In order to evaluate the overall effect of temporally dynamic selection on genotype frequency distortions, we evaluated the modelled genotype frequencies in the spat population relative to HWE estimates based on the MAF of the spat pool (figure 5). This comparison is more informative than directly relating spat genotypes to those in the fertilized egg pool because it accounts for overall changes in allele frequencies in an unbiased fashion (see the electronic supplementary material, table S2 and figure S8 comparing the two approaches). In this context, HWE represents the null hypothesis that 'final' genotype frequencies in the spat pool reflect only the relative abundance of the two alleles at that stage, irrespective of the magnitude or pattern of change in MAF. Loci with genotypic proportions in disagreement with HWE suggest an alternative hypothesis that changes in MAF were not equally accounted for across

all three genotypes owing to genetic interactions, such as dominance or overdominance.

There were *n* = 79 genotypes that significantly differed from modelled HWE frequencies (electronic supplementary material, figure S9), and the general departure from HWE across the range of MAF in the spat pool supports two complementary hypotheses to explain these distortions. For loci with rare alternative alleles (initial MAF < 20%), genotypes were skewed towards fixation of the major allele (AA up, AB/BB down). This is a hallmark signature of selection against negative or deleterious mutations (genetic load) and accounts for 35 of the 39 outlier loci (approx. 90%) in this MAF range. Loci with moderate starting MAF (greater than 20%), however, displayed a distinctly different trend whereby these distortions appeared to favour heterozygous genotypes (AB up, AA/BB down) with surprising consistency (figure 5). These departures from HWE expectations account for 28 of the 40 outlier loci (70%) in this MAF range. Heterozygote excess is traditionally interpreted as evidence of overdominance, or heterozygote advantage. The temporal patterns of genotype-specific fitness (figure 4) suggest that, in this case, 'heterozygote advantage' was characterized by reduced variability in fitness between developmental stages rather than consistently improved performance across all of them. While the net effect of this 'temporally interrupted heterotic resilience' on genotype abundance is similar to traditional examples of heterosis, the mechanisms behind it are fundamentally different.

## (d) Can temporally balanced selection be adaptive?

Balancing selection has been proposed as a mechanism that preserves genetic diversity within populations over generations, countering the effects of genetic drift and directional selection [3,6–9]. In traditional examples where balancing selection is a consequence of environmental variability, selective pressures may be heterogeneous across spatial or temporal scales and favour alternate genotypes, resulting in a balancing effect for allele frequencies in a population as a whole. In this study, we propose that the complex early life stages of oysters represent a heterogeneous developmental 'landscape' in which balancing selection opposes the effects of genetic load in a single generation. Genetic load is thought to be a driving force behind much of the larval mortality observed in oysters [34] but we find that the temporal patterns of changes in allele frequencies for a significant proportion of loci are inconsistent with simple explanations of directional selection and purging of deleterious mutations. Even if we consider the most dynamic temporal trajectories in allele frequencies (clusters 4 and 5; figures 2 and 4) to be exceptional, we still found that the majority of loci in this study (approx. 62%) had allele frequency distortions which were effectively balanced across developmental transitions (figure 3). The two most parsimonious explanations for these BD allele frequency trajectories are: (i) loci are linked to one or more genes that have contrasting consequences at different stages of development or, (ii) loci are linked to two or more genes under simultaneous but opposing (repulsive) selective pressures. The first explanation is consistent with the oscillatory [16] and stage-specific [14] patterns of gene expression observed during development in other invertebrates and the second has been demonstrated for genes involved with metamorphosis in oysters [35]; however, these two explanations are not mutually exclusive.

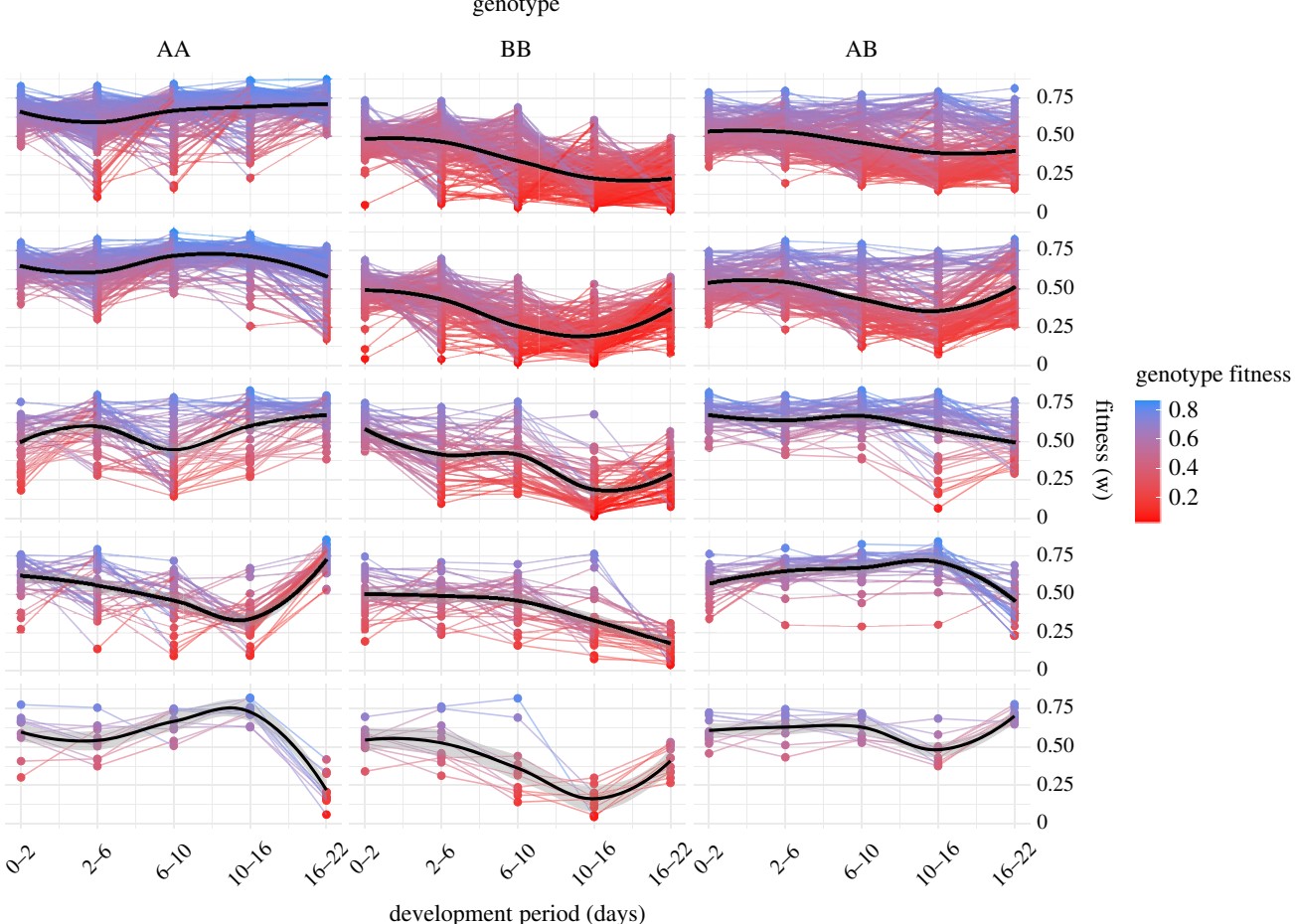

**Figure 4.** Modelled temporal patterns of genotype-specific fitness (0–1) across larval development. Each coloured line represents mean estimates of fitness for each locus from n = 50 simulations. Black lines are pooled means within each genotype and cluster (from figure 2; grey ribbons = ± s.d.). (Online version in colour.)

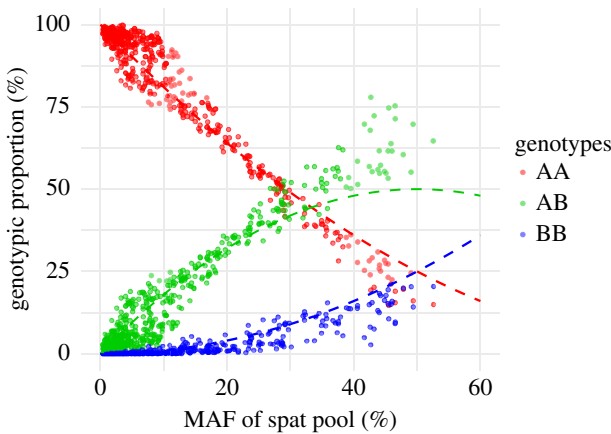

**Figure 5.** Simulated changes in genotypic proportions from fertilization to settlement. Changes in spat genotypes (circles) relative to HWE from the MAF of the spat pool (dashed lines). (Online version in colour.)

Our findings suggest that temporally balanced selective pressures may strongly influence the genetic composition of juvenile oyster populations. Temporally heterogeneous selective pressures intrinsic to larval development probably temper the effects of purifying selection for putatively negative alleles in oyster populations. Among the 751 markers examined in this study, none had minor alleles that were completely removed, or 'purged' through development (electronic supplementary material, figure S10). In this scenario, it becomes difficult to assign qualitative 'values' (i.e. negative

or deleterious) to alternate alleles which, for many loci, did not appear to be uniformly advantageous or disadvantageous to larval survival across development and may have conferred improved fitness in the heterozygous state. Furthermore, environmental stress is expected to exacerbate the lethality and dominance of negative mutations [40], but it is likely that the array of stressors impacting larval oysters in the wild (e.g. food limitation, temperature, pH) impose unique selection pressures upon various loci, alleles and genotypes. Preserving rare or 'negative' alleles in a population through temporally balanced selection may reflect an evolutionary trade-off that results in reduced survival and recruitment under optimal conditions but an improved chance of success in sub-optimal environments. In the purple sea urchin (*Strongylocentrotus purpuratus*), balancing selection and the preservation of rare variants was similarly demonstrated and proposed as an adaptive mechanism for survival in coastal environments subject to stressful environmental variables, such as ocean acidification [41]. Oysters, together with many other marine invertebrate taxa, are sedentary, highly fecund and may experience stochastic recruitment success [42] in distant and variable habitats [43,44]. It is possible that balancing selection during larval development provides long-term benefits in maintaining the genetic diversity and adaptive potential of marine invertebrates [45] at the cost of high rates of mortality within a single reproductive cohort.

In conclusion, we found that temporal changes in allele frequencies during larval development in Pacific oysters were significantly more dynamic than what would be

predicted under expectations of genetic load and directional selection. More than half of all loci analysed in this study had allele frequency trajectories which reflected balancing selection across developmental time points. Modelled estimates of genotype-specific fitness also suggest that the lethality and dominance of putatively 'negative' alleles are highly temporally variable and defy simple classification. Overall, these complex selective processes may represent an example of balancing selection that is evolutionarily beneficial to oysters and other marine invertebrate species by preserving genetic diversity and improving chances of survival in spatially and temporally variable environments.

Data accessibility. Data are available in the electronic supplementary material [46], custom scripts and codes can be found at: https://github.com/E-Durland/. All raw sequencing data (FASTQ files) are available from the Dryad Digital Repository: https://doi.org/10.5061/dryad.2z34tmpn5 [47].

Authors' contributions. E.D.: conceptualization, data curation, formal analysis, investigation, methodology, validation, visualization, writing—original draft, writing—review and editing; P.D.W.: conceptualization, supervision, validation, writing—original draft, writing—review and editing; C.L.: funding acquisition, investigation, project administration, resources, supervision, validation, writing—original draft, writing—review and editing. All authors gave final approval for publication and agreed to be held accountable for the work performed therein.

Competing interests. The authors declare no competing interests.

Funding. This report was prepared by Oregon Sea Grant under award no. NA14OAR4170064 (project no. R/SAQ-20) from the National Oceanic and Atmospheric Administration's National Sea Grant College Program, US Department of Commerce, and by appropriations made by the Oregon State Legislature. E.D. was partially funded by USDA-ARS (CRIS Project no. 2072-31000-004-00D), the Royal Swedish Academy of Sciences (KVA), as well as the HMSC Markham Fund, the National Shellfisheries Association (NSA) and the Oregon Shell Club.

Acknowledgements. The authors wish to thank Blaine Schoolfield and Matthew Gray for helping with practical aspects of larval cultivation. Sequencing was performed by the SNP&SEQ Technology Platform in Uppsala. The facility is part of the National Genomics Infrastructure (NGI) Sweden and Science for Life Laboratory. The SNP&SEQ Platform is also supported by the Swedish Research Council and the Knut and Alice Wallenberg Foundation. The statements, findings, conclusions and recommendations are those of the authors and do not necessarily reflect the views of these funders.

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
