## [Peer Review File · Proceedings of the Royal Society B: Biological Sciences]

Review History

RSPB-2020-1976.R0 (Original submission)

Review form: Reviewer 1

Recommendation

Major revision is needed (please make suggestions in comments)

Scientific importance: Is the manuscript an original and important contribution to its field?

Excellent

General interest: Is the paper of sufficient general interest?

Excellent

Quality of the paper: Is the overall quality of the paper suitable?

Good

Is the length of the paper justified?

Yes

Should the paper be seen by a specialist statistical reviewer?

No

Do you have any concerns about statistical analyses in this paper? If so, please specify them explicitly in your report.

Yes

It is a condition of publication that authors make their supporting data, code and materials available - either as supplementary material or hosted in an external repository. Please rate, if applicable, the supporting data on the following criteria.

Is it accessible?

Yes

Is it clear?

Yes

Is it adequate?

Yes

Do you have any ethical concerns with this paper?

Yes

Comments to the Author

This is a really interesting study of potentially broad significance. For a fraction of the assayed markers, the authors observed bi-directional changes of allele frequencies during the larval development, consistent with changing direction of selection. This may indicate a mechanism of balancing selection operating over the complex life cycle of the Pacific oyster. There may be some statistical issues that I mention below, but the most critical weakness is the way the study is written. In its current form, the ms is challenging to read and understand, though the overall message is not overly complicated. It's a pity, because, as written, the ms doesn't give justice to the study - it takes a lot of effort from the reader to appreciate the quality of research and follow the main line of arguments. The section "Results and discussion" is particularly difficult to follow. Problem in part may be that M&M are at the end of and it's hard to follow the R&D section before reading M&M. I suggest restructuring the ms: either move M&M before R&D, or add more information in the R&D section.

The distribution of different clusters on the linkage map suggests that they are indeed independent. It'd be useful, however, to comment on how an apparent non-independence between markers caused by linkage disequilibrium could affect your results, in particular the statistical analysis, where you consider each marker as independent. If you think it's of no consequence, please explain your reasoning.

Specific comments

l. 9-12 long, complex sentence, could be both simplified and made more informative

l. 36 maybe this sole subheading in Intro is not necessary?

l. 56 directive -> directional(?)

l. 71 "independent crosses" - please explain what you mean here, it looks to me that these crosses were not really independent in the statistical sense (though this is not crucial for the study)

l. 70-84 this paragraph is rather difficult to follow. Please try to rewrite to make it more accessible.

l. 88 what is "Type III survivorship"? If the definition is in the sentence, perhaps put the term in parentheses at the end of the sentence

l. 117-120 please provide some measure/formal test of aggregation

l. 123 say that $k = 5$ was selected as the "optimal k " using a standard procedure

l. 126-8 "The cross-classification ..." is not really clear what you mean here

Figure 3. I understand what this figure shows, but I'm not sure that it adds anything to the info already present in the text.

Figure 5 it'd be good to perform a formal statistical analysis of the results in this Figure

l. 150 - 153 "Additionally, the high level..." I really don't know what the authors want to say here

- l. 199 “method” doesn’t sound right here, perhaps “mechanism”?
- l. 297 you say that there were n=24 samples sequenced, while earlier five pools and six time points are mentioned which would give 30 samples
- l. 360-387 could you please make this section more accessible. This is an interesting approach, and it deserves to be well explained

Review form: Reviewer 2

Recommendation

Accept with minor revision (please list in comments)

Scientific importance: Is the manuscript an original and important contribution to its field?

Excellent

General interest: Is the paper of sufficient general interest?

Excellent

Quality of the paper: Is the overall quality of the paper suitable?

Good

Is the length of the paper justified?

Yes

Should the paper be seen by a specialist statistical reviewer?

No

Do you have any concerns about statistical analyses in this paper? If so, please specify them explicitly in your report.

No

It is a condition of publication that authors make their supporting data, code and materials available - either as supplementary material or hosted in an external repository. Please rate, if applicable, the supporting data on the following criteria.

Is it accessible?

No

Is it clear?

N/A

Is it adequate?

N/A

Do you have any ethical concerns with this paper?

No

Comments to the Author

The authors use Rad-Seq of pooled individuals to study allele frequency changes during development. Consistent with high mortality rates during development, the authors find several SNPs which experience dramatic frequency changes.

This is a nice, well-executed story, which will be of interest to a wider audience. My only major concern is the pitching of the manuscript. Fitness differences between developmental stages are

interesting, but I do not think that the selling point of protection of deleterious alleles fits the story well. Hence, I would recommend that the authors change their introduction to better reflect their data.

Minor suggestions:

The authors used only 5 males to start their population-is it possible that these males had "unusual" alleles that drive these patterns? In other words if a larger genetic basis would have been used for the experiment-would the results look different? It would be nice if the authors could include a discussion about this and the possibility of a few genotypes driving large allele frequency changes at many loci.

Figure 2, is the key figure of the manuscript-it would be important that the authors add the time-dependent mortality. Maybe providing the percentage of larvae surviving. This would link the death rate to observed frequency changes.

The authors should discuss that the loci with the strongest response do end up at different frequencies-which implies that either the starting allele frequencies were not at equilibrium (possible given the small genetic basis of the founders) or that later stages are also experiencing strong selection.

Is it possible that death is caused by a developmental stage specific maximum density?

Decision letter (RSPB-2020-1976.R0)

15-Sep-2020

Dear Dr Durland:

I am writing to inform you that your manuscript RSPB-2020-1976 entitled "Temporally-balanced selection pressures during development in larval oysters preserve diversity but may also shelter genetic load" has, in its current form, been rejected for publication in Proceedings B.

This action has been taken on the advice of referees, who have recommended that substantial revisions are necessary. With this in mind we would be happy to consider a resubmission, provided the comments of the referees are fully addressed. However please note that this is not a provisional acceptance.

4) Data - please see our policies on data sharing to ensure that you are complying (<https://royalsociety.org/journals/authors/author-guidelines/#data>).

Sincerely,
Professor Gary Carvalho
mailto: proceedingsb@royalsociety.org

Associate Editor
Board Member: 1
Comments to Author:
Dear Authors,

I am happy to tell that both expert reviewers found your results interesting and worthy of publishing in Proc B - after major revisions. Given that the recommended revisions are quite substantial, I believe the manuscript would require a new round of reviewing after this revision, and hence, the recommendation to reject and allow resubmission. I hope that you find the referee comments helpful, and decide to resubmit your work Proc B.

Best wishes,
Juha Merilä

Reviewer(s)' Comments to Author:
Referee: 1

Comments to the Author(s)

This is a really interesting study of potentially broad significance. For a fraction of the assayed markers, the authors observed bi-directional changes of allele frequencies during the larval development, consistent with changing direction of selection. This may indicate a mechanism of balancing selection operating over the complex life cycle of the Pacific oyster. There may be some statistical issues that I mention below, but the most critical weakness is the way the study is written. In its current form, the ms is challenging to read and understand, though the overall message is not overly complicated. It's a pity, because, as written, the ms doesn't give justice to the study - it takes a lot of effort from the reader to appreciate the quality of research and follow the main line of arguments. The section "Results and discussion" is particularly difficult to follow. Problem in part may be that M&M are at the end of and it's hard to follow the R&D section before reading M&M. I suggest restructuring the ms: either move M&M before R&D, or add more information in the R&D section.

The distribution of different clusters on the linkage map suggests that they are indeed independent. It'd be useful, however, to comment on how an apparent non-independence between markers caused by linkage disequilibrium could affect your results, in particular the statistical analysis, where you consider each marker as independent. If you think it's of no consequence, please explain your reasoning.

Specific comments

l. 9-12 long, complex sentence, could be both simplified and made more informative

l. 36 maybe this sole subheading in Intro is not necessary?

l. 56 directive -> directional(?)

l. 71 "independent crosses" - please explain what you mean here, it looks to me that these crosses were not really independent in the statistical sense (though this is not crucial for the study)

l. 70-84 this paragraph is rather difficult to follow. Please try to rewrite to make it more accessible.

l. 88 what is "Type III survivorship"? If the definition is in the sentence, perhaps put the term in parentheses at the end of the sentence

- l. 117-120 please provide some measure/formal test of aggregation
 - l. 123 say that $k = 5$ was selected as the “optimal k ” using a standard procedure
 - l. 126-8 “The cross-classification ...” is not really clear what you mean here
- Figure 3. I understand what this figure shows, but I’m not sure that it adds anything to the info already present in the text.
- Figure 5 it’d be good to perform a formal statistical analysis of the results in this Figure
- l. 150 – 153 “Additionally, the high level...” I really don’t know what the authors want to say here
 - l. 199 “method” doesn’t sound right here, perhaps “mechanism”?
 - l. 297 you say that there were $n=24$ samples sequenced, while earlier five pools and six time points are mentioned which would give 30 samples
 - l. 360-387 could you please make this section more accessible. This is an interesting approach, and it deserves to be well explained

Referee: 2

Comments to the Author(s)

The authors use Rad-Seq of pooled individuals to study allele frequency changes during development. Consistent with high mortality rates during development, the authors find several SNPs which experience dramatic frequency changes.

This is a nice, well-executed story, which will be of interest to a wider audience. My only major concern is the pitching of the manuscript. Fitness differences between developmental stages are interesting, but I do not think that the selling point of protection of deleterious alleles fits the story well. Hence, I would recommend that the authors change their introduction to better reflect their data.

Minor suggestions:

The authors used only 5 males to start their population-is it possible that these males had “unusual” alleles that drive these patterns? In other words if a larger genetic basis would have been used for the experiment-would the results look different? It would be nice if the authors could include a discussion about this and the possibility of a few genotypes driving large allele frequency changes at many loci.

Figure 2, is the key figure of the manuscript-it would be important that the authors add the time-dependent mortality. Maybe providing the percentage of larvae surviving. This would link the death rate to observed frequency changes.

The authors should discuss that the loci with the strongest response do end up at different frequencies-which implies that either the starting allele frequencies were not at equilibrium (possible given the small genetic basis of the founders) or that later stages are also experiencing strong selection.

Is it possible that death is caused by a developmental stage specific maximum density?

Author's Response to Decision Letter for (RSPB-2020-1976.R0)

See Appendix A.

RSPB-2020-3223.R1 (Revision)

Review form: Reviewer 1

Recommendation

Accept with minor revision (please list in comments)

Scientific importance: Is the manuscript an original and important contribution to its field?

Excellent

General interest: Is the paper of sufficient general interest?

Excellent

Quality of the paper: Is the overall quality of the paper suitable?

Good

Is the length of the paper justified?

Yes

Should the paper be seen by a specialist statistical reviewer?

No

Do you have any concerns about statistical analyses in this paper? If so, please specify them explicitly in your report.

No

It is a condition of publication that authors make their supporting data, code and materials available - either as supplementary material or hosted in an external repository. Please rate, if applicable, the supporting data on the following criteria.

Is it accessible?

Yes

Is it clear?

Yes

Is it adequate?

Yes

Do you have any ethical concerns with this paper?

No

Comments to the Author

The revised version is much improved, now it'd be easier for the readers to appreciate the novelty and significance of this excellent study. I have only a few minor comments.

l. 26-27 " Additionally, pleiotropy, epistasis and complex genomic architectures may also create unexpected outcomes for genes putatively affected by balancing selection, as well as alleles at linked loci which may be deleterious to the organism" Don't you think that pleiotropy, epistasis etc. can themselves be the source of balancing selection?

l. 41 and ff. I'm wondering to what extent the existing theory regarding the maintenance of variation under spatially or temporally variable selection applies to the scenario the authors describe. It'd be nice to include a brief discussion of this topic in Res & Disc.

l. 127 please provide a bit more information about the egg pool that was used to estimate the initial allele frequency. In particular, I'm wondering whether this pool can be safely considered

just a pool of fertilized eggs, or whether perhaps this was a pool of early stage developing embryos. This may potentially be important as the differences in the developmental rate at very early stages could cause severe bias in allele frequency estimates compared to zygotes (several-fold differences in the number of cells between genotypes).

l. 145-148 unclear whether you subsampled a single SNP per locus or perhaps took only loci with a single SNP

l. 180 "were in a single or uniform direction" - unclear

Decision letter (RSPB-2020-3223.R0)

02-Aug-2021

Dear Dr Durland

I am pleased to inform you that your manuscript RSPB-2020-3223 entitled "Temporally-balanced selection during development of larval Pacific oysters (*Crassostrea gigas*) inherently preserves genetic diversity within offspring" has been accepted for publication in Proceedings B.

The referee(s) have recommended publication, but also suggest some minor revisions to your manuscript. Therefore, I invite you to respond to the referee(s)' comments and revise your manuscript. Because the schedule for publication is very tight, it is a condition of publication that you submit the revised version of your manuscript within 7 days. If you do not think you will be able to meet this date please let us know.

[http://datadryad.org/submit?journalID=RSPB&manu=\(Document not available\)](http://datadryad.org/submit?journalID=RSPB&manu=(Document%20not%20available)) which will take you to your unique entry in the Dryad repository. If you have already submitted your data to dryad you can make any necessary revisions to your dataset by following the above link. Please see <https://royalsociety.org/journals/ethics-policies/data-sharing-mining/> for more details.

Sincerely,

Professor Gary Carvalho

Associate Editor

Board Member

Comments to Author:

Dear Authors,

Thank you for your revision. As you can see from the enclosed referee report, you have done good job with the revision. The additional comments from the reviewer are quite minor, and I trust you can easily accommodate these to an additional minor revision.

Best wishes,
Juha Merilä

Reviewer(s)' Comments to Author:

Referee: 1

Comments to the Author(s).

The revised version is much improved, now it'd be easier for the readers to appreciate the novelty and significance of this excellent study. I have only a few minor comments.

l. 26-27 "Additionally, pleiotropy, epistasis and complex genomic architectures may also create unexpected outcomes for genes putatively affected by balancing selection, as well as alleles at linked loci which may be deleterious to the organism" Don't you think that pleiotropy, epistasis etc. can themselves be the source of balancing selection?

l. 41 and ff. I'm wondering to what extent the existing theory regarding the maintenance of variation under spatially or temporally variable selection applies to the scenario the authors describe. It'd be nice to include a brief discussion of this topic in Res & Disc.

l. 127 please provide a bit more information about the egg pool that was used to estimate the initial allele frequency. In particular, I'm wondering whether this pool can be safely considered just a pool of fertilized eggs, or whether perhaps this was a pool of early stage developing embryos. This may potentially be important as the differences in the developmental rate at very early stages could cause severe bias in allele frequency estimates compared to zygotes (several-fold differences in the number of cells between genotypes).

l. 145-148 unclear whether you subsampled a single SNP per locus or perhaps took only loci with a single SNP

l. 180 "were in a single or uniform direction" - unclear

Author's Response to Decision Letter for (RSPB-2020-3223.R0)

See Appendix B.

Decision letter (RSPB-2020-3223.R1)

09-Aug-2021

Dear Dr Durland

I am pleased to inform you that your manuscript entitled "Temporally-balanced selection during development of larval Pacific oysters (*Crassostrea gigas*) inherently preserves genetic diversity within offspring" has been accepted for publication in Proceedings B.

Data Accessibility section

Open Access

Paper charges

Sincerely,

Appendix A

Associate Editor
Board Member: 1
Comments to Author:
Dear Authors,

I am happy to tell that both expert reviewers found your results interesting and worthy of publishing in Proc B - after major revisions. Given that the recommended revisions are quite substantial, I believe the manuscript would require a new round of reviewing after this revision, and hence, the recommendation to reject and allow resubmission. I hope that you find the referee comments helpful, and decide to resubmit your work Proc B.

Best wishes,

Juha Merilä

We thank the reviewers for the constructive feedback and pointing out areas where the clarity of the manuscript can be improved. We have re-structured the manuscript and added new analyses and information to strengthen the findings. We did not include a 'tracked changes' document with the revised version because we feel that the extent of the revisions would make such a manuscript unreadable. We hope that, after incorporating your suggestions, the new version is more comprehensive as well as approachable to readers. We have responded to specific questions and concerns below.

Reviewer(s)' Comments to Author:

Referee: 1

Comments to the Author(s)

This is a really interesting study of potentially broad significance. For a fraction of the assayed markers, the authors observed bi-directional changes of allele frequencies during the larval development, consistent with changing direction of selection. This may indicate a mechanism of balancing selection operating over the complex life cycle of the Pacific oyster. There may be some statistical issues that I mention below, but the most critical weakness is the way the study is written. In its current form, the ms is challenging to read and understand, though the overall message is not overly complicated. It's a pity, because, as written, the ms doesn't give justice to the study – it takes a lot of effort from the reader to appreciate the quality of research and follow the main line of arguments. The section "Results and discussion" is particularly difficult to follow. Problem in part may be that M&M are at the end of and it's hard to follow the R&D section before reading M&M. I suggest restructuring the ms: either move M&M before R&D, or add more information in the R&D section.

We agree and have modified the MS accordingly

The distribution of different clusters on the linkage map suggests that they are indeed independent. It'd be useful, however, to comment on how an apparent non-independence between markers caused by linkage disequilibrium could affect your results, in particular the statistical analysis, where you consider each marker as independent. If you think it's of no consequence, please explain your reasoning.

We agree that this point needed some clarification. We have now included analyses and figures in the supplement (files S2-8) to more thoroughly address this. We have also modified the methods (lines 194-208) to describe the mapping methods. We discuss this in the context of the results on lines 324-330. In this case we assess independence of markers not only as a factor of genomic distance (cM) but also the number of recombinations present in the population. Since the starting pool of larvae was $n \approx 10^5$, we expect there to be extensive recombination between even moderately close markers (e.g. ~10 cM) in the population as a whole.

Specific comments

I. 9-12 long, complex sentence, could be both simplified and made more informative

Remedied

I. 36 maybe this sole subheading in Intro is not necessary?

Agreed, removed

I. 56 directive -> directional(?)

Remedied

I. 71 "independent crosses" – please explain what you mean here, it looks to me that these crosses were not really independent in the statistical sense (though this is not crucial for the study)

We now refer to these as 'independently fertilized crosses'

I. 70-84 this paragraph is rather difficult to follow. Please try to rewrite to make it more accessible.

We have restructured this paragraph (lines 68-80) to be more readable.

I. 88 what is "Type III survivorship"? If the definition is in the sentence, perhaps put the term in parentheses at the end of the sentence

Remedied (lines 268-270)

I. 117-120 please provide some measure/formal test of aggregation

We have added this, please see major comment above

I. 123 say that $k = 5$ was selected as the "optimal k " using a standard procedure

Done (lines 187-189)

I. 126-8 “The cross-classification ...” is not really clear what you mean here

Fixed (lines 287-291)

Figure 3. I understand what this figure shows, but I’m not sure that it adds anything to the info already present in the text.

We feel that this figure adds to the information from the text (ephemeral vs overall change) by presenting this finding in the context of the scale of allele frequency change from day 2 to 22 (which is not represented elsewhere).

Figure 5 it’d be good to perform a formal statistical analysis of the results in this Figure

We have now included this. New text can be found in the methods (lines 254-264), a revised discussion (section beginning line 365) and supplemental figures S8 and S9.

I. 150 – 153 “Additionally, the high level...” I really don’t know what the authors want to say here

We have reformatted this section for improved accessibility to the readers (from line 209)

I. 199 “method” doesn’t sound right here, perhaps “mechanism”?

Agreed, fixed.

I. 297 you say that there were n=24 samples sequenced, while earlier five pools and six time points are mentioned which would give 30 samples

We have clarified the samples and data structure on lines 148-150

I. 360-387 could you please make this section more accessible. This is an interesting approach, and it deserves to be well explained

Thank you for the encouragement and suggestion. We have re-formatted this explanation (starting line 209) and we hope it is improved. We have also supplemented this with a full demonstration and explanation on the github page: https://github.com/E-Durland/Genotype_forecaster

Referee: 2

Comments to the Author(s)

The authors use Rad-Seq of pooled individuals to study allele frequency changes during development. Consistent with high mortality rates during development, the authors find several SNPs which experience dramatic frequency changes.

This is a nice, well-executed story, which will be of interest to a wider audience. My only major concern is the pitching of the manuscript. Fitness differences between developmental stages are interesting, but I do not think that the selling point of protection of deleterious alleles fits the

story well. Hence, I would recommend that the authors change their introduction to better reflect their data.

We thank the reviewer for this suggestion and appreciate that the structure of this dataset is not ideally suited for identifying the detailed mechanisms of genetic load. For the Pacific oyster, however, the effects of genetic load during larval development is arguably the most important mechanism underpinning overall recruitment success as well as population genetic parameters^{1, 2}. Given this, and the fact that this dataset represents the most temporally resolved and marker rich to date, we feel that we cannot avoid this as an important aspect of the interpretation. We agree that this dataset cannot concretely confirm that genetic load is sheltered through balancing selection. Given that the majority of loci do have balanced allele frequency trajectories and that we see no fixation of alleles (Figure S10) provides reasonably strong evidence that the two observations are not independent. Furthermore, the patterns of balancing selection we demonstrate in this paper provide a novel and important way to explain how genetic load in this species is so persistent even in the face of intense viability selection.

Overall, we have tried to broaden the scope of the paper to contextualize our findings in multiple ways. We feel that genetic load has such an overriding importance for this species that we cannot neglect the inferences that our results have for understanding its role in oyster populations.

Minor suggestions:

The authors used only 5 males to start their population-is it possible that these males had "unusual" alleles that drive these patterns? In other words if a larger genetic basis would have been used for the experiment-would the results look different? It would be nice if the authors could include a discussion about this and the possibility of a few genotypes driving large allele frequency changes at many loci.

This is a good point, we have now discussed this in the paragraph starting on Line 295.

Figure 2, is the key figure of the manuscript-it would be important that the authors add the time-dependent mortality. Maybe providing the percentage of larvae surviving. This would link the death rate to observed frequency changes.

We have now included both the time-dependent mortality and the change in mortality to this figure

The authors should discuss that the loci with the strongest response do end up at different frequencies-which implies that either the starting allele frequencies were not at equilibrium (possible given the small genetic basis of the founders) or that later stages are also experiencing strong selection.

While we can fully expect that there are additional selection pressures on oyster post-metamorphosis, the genetic components of the two periods of the life cycle appear to be largely independent³. Additionally, given that the vast majority of mortality occurs during larval development (typically >90%, and ~96% in this experiment) the power of selection in juvenile and adult stages is dwarfed by those occurring early in the life cycle. With regards to

equilibrium of alleles, expectations from previous studies are that negative alleles, indeed, are not in equilibrium in naturalized oyster populations owing to the rapid rate of mutation in this species. We have now included these concepts in the discussion starting at line 295.

Is it possible that death is caused by a developmental stage specific maximum density?

We controlled for density of the larvae throughout the experiment with scheduled reductions at days: 2, 6 and 14 (line 109). In our experience these are very conservative densities (compared to commercial hatchery practices) and accommodate the highest performance (survival and growth) possible.

References:

1. Plough LV, Hedgecock D. Quantitative trait locus analysis of stage-specific inbreeding depression in the Pacific oyster *Crassostrea gigas*. *Genetics* **189**, 1473-1486 (2011).
2. Plough LV, Shin G, Hedgecock D. Genetic inviability is a major driver of type III survivorship in experimental families of a highly fecund marine bivalve. *Molecular Ecology* **25**, 895-910 (2016).
3. Ernande B, Clobert J, McCombie H, Boudry P. Genetic polymorphism and trade-offs in the early life-history strategy of the Pacific oyster, *Crassostrea gigas* (Thunberg, 1795): a quantitative genetic study. *J Evol Biol* **16**, 399-414 (2003).

Appendix B

Response to reviewers:

Comments to the Author(s).

The revised version is much improved, now it'd be easier for the readers to appreciate the novelty and significance of this excellent study. I have only a few minor comments.

We thank the reviewer for their support and constructive feedback through this process. We have responded to the comments below and addressed these sections of the manuscript accordingly.

I. 26-27 “Additionally, pleiotropy, epistasis and complex genomic architectures may also create unexpected outcomes for genes putatively affected by balancing selection, as well as alleles at linked loci which may be deleterious to the organism” Don't you think that pleiotropy, epistasis etc. can themselves be the source of balancing selection?

We agree with the reviewer and have modified the text in the introduction to more clearly acknowledge that these complexities are not merely analytically confusing but biologically meaningful!

I. 41 and ff. I'm wondering to what extent the existing theory regarding the maintenance of variation under spatially or temporally variable selection applies to the scenario the authors describe. It'd be nice to include a brief discussion of this topic in Res & Disc.

We propose that the scenario we describe is very comparable to temporally variable selection in populations across generations, but taking place across developmental stages within a single cohort of oyster larvae. We have fortified the link between these two concepts in the discussion starting on line 399.

I. 127 please provide a bit more information about the egg pool that was used to estimate the initial allele frequency. In particular, I'm wondering whether this pool can be safely considered just a pool of fertilized eggs, or whether perhaps this was a pool of early stage developing embryos. This may potentially be important as the differences in the developmental rate at very early stages could cause severe bias in allele frequency estimates compared to zygotes (several-fold differences in the number of cells between genotypes).

This is a good point. We obtained samples of eggs after validating that that fertilization was effective, represented by the presence of a polar body, approximately ~1hr post fertilization. We have now clarified this in the text. While cell division may have varied among the 95 crosses and thereby skewed estimates of the true allele frequency ratio in the pool as a whole, this artifact is not likely to significantly persist past embryogenesis when larval size becomes relatively uniform. Additionally, we found, like previously reported by Plough and Hedgecock (2011), that initial development (0-2 days) had the least amount of genetic change of all the stages (Figure 1 in this study). Consequentially, we feel that any such bias in our method at the initial sampling is unlikely to affect the overall trajectory of allele frequencies that we document in these oysters across developmental stages.

I. 145-148 unclear whether you subsampled a single SNP per locus or perhaps took only loci with a single SNP

Another good point deserving clarification. We have now included the following text at line 151: “(i.e. tags with >1 SNP were excluded)”

I. 180 “were in a single or uniform direction” – unclear

We have changed this to: “ ‘Uni-directional’ changes (UD) were identified by significant changes in allele frequencies between consecutive time points (e.g. day 2 to day 6) which were solitary or uniform in direction (up or down) “